# Cystic fibrosis risk variants confer protection against inflammatory bowel disease

## Graphical abstract

## Authors

Mingrui Yu, Qian Zhang, Kai Yuan, ...,
Carl A. Anderson, Mark J. Daly,
Hailiang Huang

## Correspondence

ca3@sanger.ac.uk (C.A.A.),
mjdaly@broadinstitute.org (M.J.D.),
hhuang@broadinstitute.org (H.H.)

## In brief

Through large-scale sequencing analysis, Yu et al. found that cystic fibrosis risk variants reduce susceptibility to IBD. This finding opens the possibility of developing selective, tissue-targeted CFTR modulators as a novel IBD therapeutic intervention. Their analysis also highlights the critical and unmet need for effective variant prioritization in gene-based burden tests.

## Highlights

- Large-scale sequencing study of inflammatory bowel disease (IBD)

- Genetic mutations known to cause cystic fibrosis are protective against IBD

- Accurate variant functional annotation enhances the power of rare-variant burden tests

- *In silico* variant annotations (e.g., AlphaMissense) underperform clinical annotations

 Yu et al., 2026, Cell Genomics 6, 101071
February 11, 2026 © 2025 The Authors. Published by Elsevier Inc.

**CellPress**

## Article

# Cystic fibrosis risk variants confer protection against inflammatory bowel disease

Mingrui Yu,[1,2,3] Qian Zhang,[4,5,6] Kai Yuan,[1,2,3,6] Aleksejs Sazonovs,[4] Christine R. Stevens,[1,2,3,6] Laura Fachal,[4,5,6] International Inflammatory Bowel Disease Genetics Consortium Sequencing Group,[12] Christopher A. Lamb,[5,6,7,8,9] Carl A. Anderson,[4,5,6,*] Mark J. Daly,[1,2,3,6,10,11,*] and Hailiang Huang[1,2,3,6,11,13,*]

[1]Program in Medical and Population Genetics, The Broad Institute of MIT and Harvard, Cambridge, MA, USA
[2]Stanley Center for Psychiatric Research, the Broad Institute of MIT and Harvard, Cambridge, MA, USA
[3]Analytic and Translational Genetics Unit, Department of Medicine, Massachusetts General Hospital, Boston, MA, USA
[4]Genomics of Inflammation and Immunity Group, Human Genetics Programme, Wellcome Sanger Institute, Wellcome Genome Campus, Hinxton, Cambridgeshire, UK
[5]UK Inflammatory Bowel Disease Genetics Consortium, Cambridge, UK
[6]International Inflammatory Bowel Disease Genetics Consortium
[7]The Newcastle upon Tyne Hospitals NHS Foundation Trust, Newcastle upon Tyne, UK
[8]Translational & Clinical Research Institute, Newcastle University, Newcastle Upon Tyne, UK
[9]NIHR IBD BioResource
[10]Institute for Molecular Medicine Finland, FIMM, HiLIFE, University of Helsinki, Helsinki, Finland
[11]NIDDK Inflammatory Bowel Disease Genetics Consortium
[12]Further details can be found in the supplemental information
[13]Lead contact
*Correspondence: ca3@sanger.ac.uk (C.A.A.), mjdaly@broadinstitute.org (M.J.D.), hhuang@broadinstitute.org (H.H.)

## SUMMARY

Genetic mutations that yield a defective cystic fibrosis (CF) transmembrane regulator (CFTR) protein cause CF, a life-limiting autosomal-recessive Mendelian disorder. A protective role of *CFTR* loss-of-function mutations in inflammatory bowel disease (IBD) has been suggested, but its evidence has been inconclusive and contradictory. Here, leveraging a large IBD exome sequencing dataset comprising 38,558 cases and 66,945 controls of European ancestry in the discovery stage and a combined total of 42,475 cases and 192,050 controls across diverse ancestry groups in the replication stage, we established a protective role of CF-risk variants against IBD based on the association test of *CFTR* deltaF508 ($p$ = 8.96E−11) and the gene-based burden test of CF-risk variants ($p$ = 3.9E−07). Furthermore, we assessed variant prioritization methods, including AlphaMissense, using clinically annotated CF-risk variants as the gold standard. Our findings highlight the critical and unmet need for effective variant prioritization in gene-based burden tests.

## INTRODUCTION

Genetic mutations that yield defective cystic fibrosis transmembrane regulator (CFTR) protein are known to cause cystic fibrosis (CF),[1] one of the most common life-limiting autosomal-recessive genetic disorders among individuals of European ancestry. *CFTR* is a transmembrane chloride ion channel gene highly expressed on the apical surface of epithelial cells.[2] The molecular and cellular functions of *CFTR* have been well studied and include a principal role for active chloride and bicarbonate secretion by epithelial cells.[3–6]

In addition to its role in the lungs, *CFTR* also influences the physiology of the gastrointestinal (GI) tract. In the GI tract, ion secretion and the resultant osmotic gradient regulate pH, mucous hydration, and luminal water content, in turn impacting stool consistency and frequency.[7] This mechanism is exemplified in cholera (*Vibrio cholerae*) infection, where people may experience profuse secretory diarrhea through binding of

cholera toxin to intestinal epithelial cells, leading to overstimulation of CFTR via activation of adenylate cyclase and an increase in cyclic adenosine monophosphate (cAMP).[8] Harnessing a pro-secretory mechanism, linaclotide and lubiprostone are treatments for chronic constipation[9,10]; linaclotide, a peptide agonist of the guanylate cyclase C receptor, increases intra-epithelial cell cyclic guanosine monophosphate (cGMP), leading to activation of CFTR[11]; and lubiprostone, an activator of ClC-2 chloride channels on the apical surface of intestinal epithelial cells, also indirectly activates CFTR via prostanoid receptor signaling.[12]

The impact of homozygous loss-of-function *CFTR* mutations on intestinal mucus function has been well characterized in CF, a condition where constipation is a common clinical feature.[13] Studies have shown that, in naive colon cells, the deltaF508 CFTR protein does not fully mature or reach the cell surface properly, which prevents it from helping move chloride ions out of the cells.[14] The loss of CFTR-mediated bicarbonate secretion

**Table 1. Study sample sizes**

| Center | Ancestry | Stage | Method | Control | CD | UC | IBD-U |
|--------|----------|-------|--------|---------|-----|-----|-------|
| Broad | EUR | discovery | WES | 66,945 | 21,478 | 14,353 | 2,727 |
| Broad | AFR.AMR | replication | WES | 9,976 | 2,553 | 1,664 | 204 |
| Broad | EAS | replication | WES | 1,462 | 1,386 | 199 | 39 |
| Broad | SAS | replication | WES | 670 | 247 | 338 | 48 |
| Sanger | EUR | replication | WES | 168,100 | 10,722 | 13,147 | 6,211 |
| Sanger | EUR | replication | WGS | 11,842 | 5,717 | 0 | 0 |

The number of study subjects in each analysis subset, post QC (STAR Methods). WES, whole-exome sequencing; WGS, whole-genome sequencing.

results in hyperviscous, poorly hydrated mucus in the epithelium, followed by inflammation and dysfunction of the GI tract.[15–17]

The effect of heterozygous *CFTR* mutations on gut physiology, however, is not fully clear. A major GI disorder that also involves mucus barrier disruption and inflammation is inflammatory bowel disease (IBD). IBD is a class of complex polygenic disorders with chronic inflammation in the GI tract. IBD has two subtypes: Crohn's disease (CD) and ulcerative colitis (UC). Genome-wide association studies have identified over 200 independent genomic loci associated with IBD.[18–22] Many of these loci implicate genes involved in innate and adaptive immune regulation, epithelial barrier function, and host-microbe interactions.[23] While most findings have focused on risk alleles, several genetic variants have been shown to confer protection against IBD. For example, the *IL23R* Arg381Gln missense mutation protects against both CD and UC (odds ratio = 0.38) by disrupting the interleukin-23 (IL-23) receptor signaling pathway.[24,25] A rare splice site variant in *CARD9* that leads to premature truncation of the transcript and acts in a dominant-negative manner for CARD9-mediated cytokine production also confers protection (odds ratio = 0.29).[26,27]

Unlike risk alleles, which highlight pathways where dysfunction promotes disease, protective alleles provide direct evidence that modulation in the opposite direction may be beneficial and tolerated in humans. This has already informed successful therapies for other complex diseases; for example, individuals with loss-of-function *PCSK9* mutations have been found to have lifelong low levels of low-density lipoprotein (LDL) cholesterol and reduced risk of coronary artery disease, which inspired the development of PCSK9 inhibitors that lower LDL and are now widely used for hypercholesterolemia and cardiovascular risk reduction. Together, risk and protective genetic mutations can provide complementary insights into IBD pathogenesis.

Earlier research has suggested that loss-of-function variants in the *CFTR* gene may lead to a lower prevalence of intestinal inflammation[28] and potentially protect against IBD. However, such evidence has been inconsistent, with studies showing conflicting results—some linking these variants to increased risk, others suggesting protection, and some finding no effect at all.[29–32] Here, leveraging a large IBD exome sequencing dataset comprising 38,558 cases and 66,945 controls of European ancestry in the discovery stage and a combined total of 42,475 cases and 192,050 controls across diverse ancestry groups in the replication stage, we provide strong genetic evidence, using single-variant and gene-based burden tests, that CF-risk variants confer protection against IBD.

## RESULTS

### Study subjects

Exome sequencing was performed at the Broad Institute. Study subjects were recruited from different centers and shared with the International Inflammatory Bowel Disease Genetics Consortium (IIBDGC) for 45,236 IBD cases and 79,053 controls after quality control (Tables 1 and S1). Each subject was then assigned to one of four ancestry groups before analysis (Figure 1): European (EUR), African American (AFR.AMR), East Asian (EAS), and Southeast Asian (SAS). Additional exome and whole-genome sequencing were performed at the Sanger Institute. Exome sequencing was performed on 10,722 CD patients, 13,147 UC patients, and 6,211 IBD unclassified (IBD-U) patients, which were matched with exome sequencing data obtained from 168,100 IBD-free subjects as controls from the UK Biobank. Extensive quality control was undertaken to harmonize the case and control exome sequencing data (STAR Methods). Whole-genome sequencing was performed on another independent sample of 5,717 CD and 11,842 controls. All subjects in the Sanger dataset are of European ancestry. Details on sequencing and data quality-control protocols are described in STAR Methods and Figure S1.

### CF-causing variant deltaF508 confers a protective effect against IBD

The *CFTR* deltaF508 variant is an in-frame deletion commonly observed in the European population, with a minor-allele frequency (MAF) of 1.5% in non-Finnish Europeans (chr7:117559590:ATCT:A in Genome Reference Consortium Human Build 38 [GRCh38]). Homozygous deltaF508 has full penetrance for CF and is the most common CF-causing variant. We tested the association between deltaF508 and IBD, including its subtypes, using a logistic mixed model for both the Broad and Sanger EUR studies, followed by meta-analysis (STAR Methods; Table 2). We found exome-wide significant associations (threshold defined as 1E−6) for IBD ($p$ = 8.96E−11, odds ratio [OR] [95% confidence interval (CI)] = 0.82 [0.77–0.87]) and its subtype, CD ($p$ = 5.49E−9, OR [95% CI] = 0.79 [0.73–0.85]). The association between deltaF508 and UC ($p$ = 9.95E−4, OR [95% CI] = 0.87 [0.79–0.94]) is nominally significant (threshold defined as 0.05), likely due to its smaller effect size and sample size. Reassuringly, all constituent studies report a consistent protective effect. In addition, there are no known IBD-associated variants (defined as variants with posterior inclusion probability >5% from fine mapping[33] or reported in the published sequencing and genome-wide

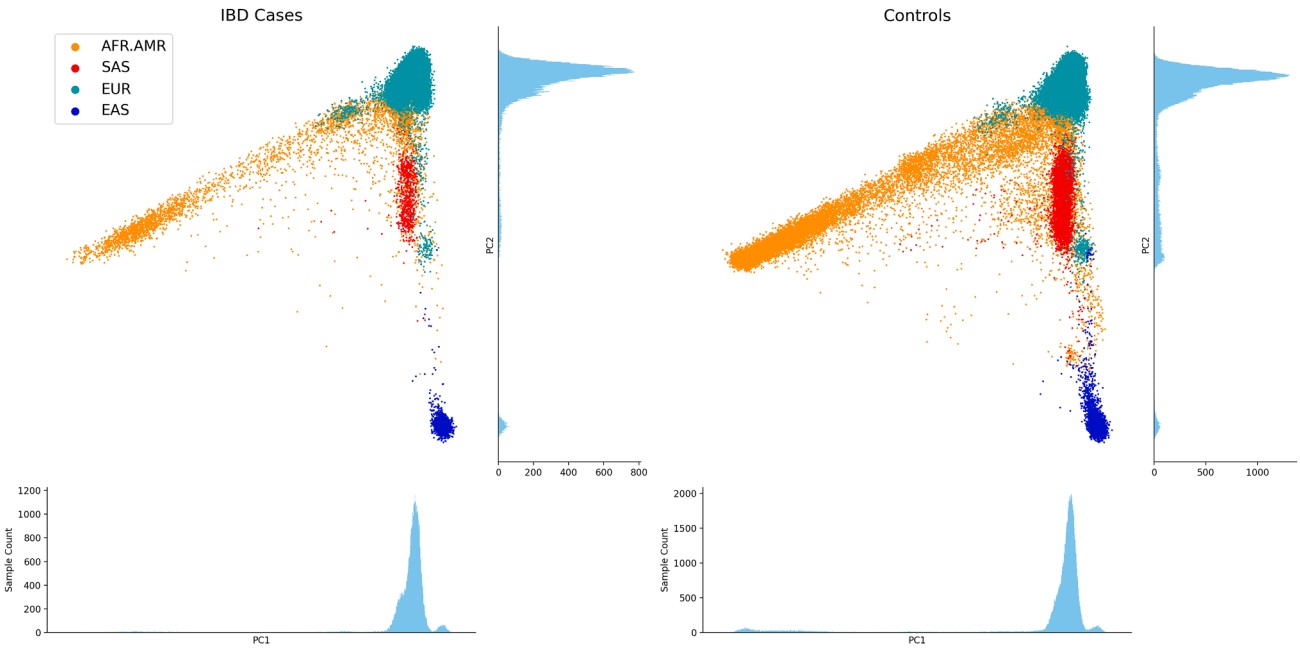

**Figure 1. Ancestry assignment in the Broad studies**
Subjects are projected onto the principal component (PC) space calculated from subjects from 1000 Genomes Project Phase 3 (STAR Methods). Predicted genetic ancestries are indicated by their colors. IBD case (left) and control (right) subjects are plotted separately. Sample distributions along the PC1 and PC2 axes are shown in the histograms (number of bins = 1,000).

association studies[21,26,34]) within 1 million base pairs of deltaF508. Therefore, it is highly unlikely that the deltaF508 association tags a known IBD genetic association.

To limit the potential for CF status to bias the allele frequency estimates in our cases and controls, we repeated the association analysis after excluding 4 cases and 47 controls who were homozygous carriers of deltaF508 in the discovery and replication samples combined as well as 37 cases and 170 controls deemed potential compound heterozygous carriers (Table S2), defined as individuals with two or more variants annotated as "CF-causing" or "Varying clinical consequence" in the Clinical and Functional Translation of *CFTR*[35] (CFTR2) database. Our conclusion remained valid after excluding these individuals; the associations between *CFTR* deltaF508 and IBD after meta-analysis remain significant exome wide ($p$ = 1.16E−08, OR [95% CI] = 0.84 [0.78–0.88]; Table S3). Since no known and credible causes of CF exist aside from inheriting two copies of the defective CFTR gene, and the CFTR2 database provides nearly complete coverage of CF causal variants with population frequency observable and meaningful in this study, our approach—using genetic data to identify CF patients—is sufficient to ensure the validity of our conclusion.

### CF-causing variants confer protection against IBD in gene-based burden tests

The CFTR2 database contains a comprehensive list of *CFTR* variants curated for their causality of CF. The curation process, performed by the CFTR2 team, included reviewing the clinical symptoms of carriers and evaluating how the variant changes the CFTR protein. The Broad discovery dataset captured 1036

*CFTR* variants (Table S4). 170 of the 1,036 *CFTR* variants were annotated in CFTR2 as "CF-causing" (109), "Varying clinical consequence" (33), "Non CF-causing" (19), or "Unknown significance" (9). The remaining 866 variants did not have an annotation in CFTR2. We defined "CF-risk" variants (142) as variants annotated as "CF-causing" or "Varying clinical consequences" in CFTR2, and "Non-CF-risk" variants (893) as variants annotated as "Non CF-causing," "Variants of unknown significance" in CFTR2, or not annotated in CFTR2. We chose this approach because CFTR2 provides nearly complete coverage of variants known to cause CF; therefore, variants not annotated in CFTR2 are far more likely to be non-CF causing than CF causing undetected. "CF-risk" variants represent a subset of the missense variants curated in CFTR2 to be pathogenic for CF. "Non-CF-risk," on the contrary, are missense variants that were either curated to be not pathogenic or have insufficient data for curation. We include both classes of variants in the testing, with the "non-CF-risk" variants as a negative control.

While no variants other than deltaF508 reached significance of $p < 0.01$ (before multiple testing correction) in our single-variant tests, CF-risk variants are predominantly protective against IBD ($p$ = 0.002, binomial test using variants with minor allele count ≥10 in the Broad EUR study; Figure 2A). Conversely, non-CF-risk variants were not enriched with protective effects, which was expected for the negative control variant set ($p$ = 0.35, binomial test using variants with minor allele count ≥10 in the Broad EUR study; Figure 2B).

Based on these observations, we hypothesize that variants impairing *CFTR* function, inferred from their CF causality, are protective against IBD. To test this hypothesis, we did an

**Table 2. Association between deltaF508 and CD, UC, and IBD**

| Study | Phenotype | MAF (case) | MAF(control) | *p* value | OR | 95% CI |
|-------|-----------|-----------|--------------|-----------|-----|--------|
| Broad WES (discovery) | CD | 0.0097 | 0.0136 | 1.53E−07 | 0.74 | 0.65–0.82 |
| | UC | 0.0096 | 0.0136 | 3.35E−04 | 0.79 | 0.69–0.90 |
| | IBD | 0.0097 | 0.0136 | 8.50E−10 | 0.75 | 0.68–0.82 |
| Sanger WES (replication) | CD | 0.0135 | 0.0161 | 3.52E−02 | 0.87 | 0.77–0.99 |
| | UC | 0.0141 | 0.0161 | 2.04E−01 | 0.93 | 0.83–1.04 |
| | IBD | 0.0137 | 0.0161 | 1.47E−02 | 0.90 | 0.82–0.98 |
| Sanger WGS (replication) | CD/IBD[a] | 0.0114 | 0.0157 | 8.32E−03 | 0.73 | 0.58–0.92 |
| Meta-analysis | CD | – | – | 5.49E−09 | 0.79 | 0.73–0.85 |
| | UC | – | – | 9.95E−04 | 0.87 | 0.79–0.94 |
| | IBD | – | – | 8.96E−11 | 0.82 | 0.77–0.87 |

MAF, minor-allele frequency; OR, odds ratio; CI, confidence interval.
[a]Sanger WGS only included CD patients.

unweighted burden test of *CFTR* using the 141 CF-risk variants (deltaF508 not included, as, on its own, it confers a significant protective effect; clinical_annotation(CFTR2) column in Table S4). We found that these variants, in aggregation, have a significant protective effect on IBD in the Broad EUR study (*p* = 4.3E−06, OR [95% CI] = 0.81 [0.74–0.89]) (Table 3), which was replicated in the Sanger whole-exome sequencing (WES) dataset at nominal significance (*p* = 0.03, OR [95% CI] = 0.90 [0.81–0.99]) and in the Sanger whole-genome sequencing (WGS) dataset for the direction of effect (*p* = 0.07, OR [95% CI] = 0.85 [0.72–1.01]). The Sanger WGS is much smaller in sample size; thus, the lack of significance is likely due to its limited power (power to replicate at nominal significance is only 41%; STAR Methods). Combining all three datasets using the fixed-effect meta-analysis, we found that CF-risk variants in *CFTR* collectively confer a protective effect of OR [95% CI] = 0.85 [0.80–0.90] with *p* = 3.9E−7. This association was nominally significant when tested for CD and UC separately (CD: *p* = 8.8E−04, OR [95% CI] = 0.88 [0.81–0.95]; UC: *p* = 3.6E−06, OR [95% CI] = 0.80 [0.72–0.87]) (Table S5). On the contrary, despite a similar composite allele frequency (CAF), we did not observe a significant association with IBD (Table 3) using the non-synonymous variants that were classified as non-CF-risk in *CFTR* (most_severe_consequence and clinical_annotation(CFTR2) columns in Table S4) that had MAF < 0.1% (for selecting rare variants not in linkage disequilibrium [LD] and more likely to have a strong causal effect). This is expected as for the negative control, reinforcing our hypothesis that mutations affecting *CFTR* protein function, inferred from their CF causality, are protective against IBD.

**Variant annotation plays a critical role in burden tests**
The power of rare-variant burden tests increases with the composite allele frequency and the ratio of causal to non-causal variants included in the analysis. Therefore, identifying a variant set that maximizes the inclusion of causal variants while minimizing non-causal variants is crucial for the power of the burden test. CF-risk variants defined in CFTR2 were an effective variant set for this purpose. Unfortunately, very few genes are as thoroughly studied and clinically annotated as *CFTR*. Most human genes

lack accurate variant effect annotation. Molecular or *in silico* variant pathogenicity predictions can play an important role in systematically annotating variants to enhance the power of rare variant burden tests. Recently, AlphaMissense (AM),[36] a machine learning model trained on protein structure prediction and evolutionary constraint information for predicting missense variant pathogenicity, has been shown to have more accurate missense pathogenicity predictions, as evidenced by better correlations with functional assays than previous prediction models.[37] Here, we evaluated the utility of AM, along with commonly used molecular function annotations, for prioritizing *CFTR* missense variants for inclusion in burden tests.

A common approach employed by the burden test is to use predicted loss-of-function variants or non-synonymous variants with an MAF cutoff (<0.1%) to select rare variants not in LD and more likely to have a strong causal effect. In both scenarios, we found much weaker evidence of association (*p* = 0.042 and 1.01E−03, respectively; Table 4) compared to the burden test using CF-risk variants. This demonstrates that clinical variant annotations from CFTR2 outperform naive molecular consequence annotations in characterizing the impact of rare *CFTR* variants on IBD risk.

We then assigned the AM scores for 698 *CFTR* missense variants available in the Broad EUR study, among which 74 are CF-risk variants according to the CFTR2 database (STAR Methods). We observed that most CF-risk variants were predicted as "likely pathogenic" (AM score >0.56) (Figure 3A), and those with particularly high scores (AM score >0.8) exhibited stronger protective effects against IBD (Figure 3B). However, among "likely pathogenic" variants by AM, 73% were either classified as "Unknown significance"/"Non CF-causing" (3%) or not present in CFTR2 (70%), suggesting that AM has a low specificity. Variants not present in CFTR2 are also very likely to be non-risk variants because CFTR2 provides nearly complete coverage of CF causal variants. This is supported by the observation that the proportion of "likely pathogenic" variants was statistically indistinguishable between the "Non CF-causing/Unknown significance" group and the "not present in CFTR2" group (25% vs. 26%, *p* = 0.64). Both proportions were significantly lower than that of CF-risk variants (84%, *p* < 2.2E−16), supporting the grouping of variants

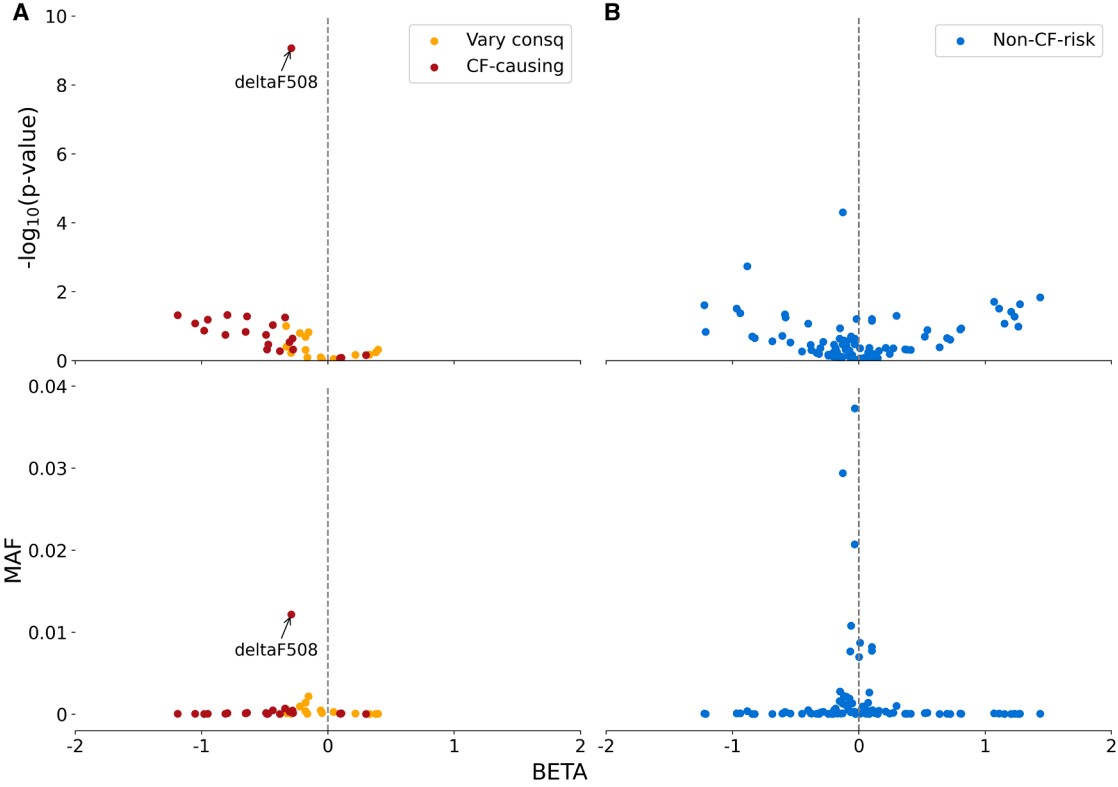

**Figure 2. Single-variant association test for _CFTR_ variants in the Broad EUR dataset**
(A) Distribution of _p_ values (from logistic mixed model) and MAF for 37 CF-risk variants with minor allele count (MAC) ≥ 10.
(B) Distribution of _p_ values (from logistic mixed model) and MAF for 96 non-CF-risk variants with MAC ≥ 10.

absent from CFTR2 with those classified as "Unknown significance"/"Non CF-causing."

The power to detect association via the burden test can be influenced by the AM score threshold used for selecting variants. To investigate this, we performed burden tests by selecting _CFTR_ missense variants (restricted to MAF <0.1%) with the AM score threshold ranging from 0 to 1 in increments of 0.01. Our analysis showed that a higher AM score threshold does

not always improve the statistical power of the burden test. Statistical power is sensitive to both the effect size (increases with the AM score threshold; Figure 3B) and the CAF (decreases with the AM score threshold; Figure 3C). Specifically, within the score range of 0.56–1, the burden test significance decreased with more stringent score cutoffs due to a reduction in CAF despite a stronger effect size (Figure 3D). Even at the optimal AM score cutoff, the significance of the association using the

**Table 3. _CFTR_ burden test using clinical annotations**

| Study | Phenotype | Molecular annotation | Clinical annotation | MAF | No. of variants | CAF[a] | _p_ value | OR | OR (95% CI) |
|---|---|---|---|---|---|---|---|---|---|
| Broad WES | IBD | – | CF-risk | – | 141 | 0.012 | 4.3E−06 | 0.81 | 0.74–0.89 |
| Sanger WES | IBD | – | CF-risk | – | 128 | 0.011 | 0.03 | 0.90 | 0.81–0.99 |
| Sanger WGS | CD/IBD[b] | – | CF-risk | – | 56 | 0.023 | 0.07 | 0.85 | 0.72–1.01 |
| Meta-analysis | IBD | – | CF-risk | – | – | – | 3.9E−07 | 0.85 | 0.80–0.90 |
| Broad WES | IBD | non-synonymous | non-CF-risk | <0.001 | 697 | 0.017 | 0.37 | 0.97 | 0.90–1.05 |
| Sanger WES | IBD | non-synonymous | non-CF-risk | <0.001 | 844 | 0.021 | 0.09 | 0.94 | 0.87–1.01 |
| Sanger WGS | CD/IBD[b] | non-synonymous | non-CF-risk | <0.001 | 180 | 0.020 | 0.55 | 0.94 | 0.78–1.14 |
| Meta-analysis | IBD | – | non-CF-risk | – | – | – | 0.06 | 0.95 | 0.91–1.00 |

[a]Defined as the frequency of observing carriers of at least one variant of interest in the study. We applied an MAF upper bound to burden tests using non-CF-risk variants so that these variants are less likely in LD, more likely to have a strong causal effect, and have similar CAFs to those of CF-risk variants.
[b]Sanger WGS only included CD patients.

**Table 4. *CFTR* burden test using molecular annotations**

| Molecular annotation | MAF | Number of variants | CAF | *p* value | OR | 95% CI |
|---|---|---|---|---|---|---|
| Predicted loss-of-function | <0.001 | 93 | 0.003 | 0.042 | 0.83 | 0.69–0.99 |
| Non-synonymous | <0.001 | 782 | 0.023 | 1.01E−03 | 0.90 | 0.85–0.96 |

Definitions of predicted loss-of-function and non-synonymous variants are described in STAR Methods.

AM-defined gene set is orders of magnitude worse than the clinically defined gene set from CFTR2, suggesting that further improvements of *in silico* variant annotations, such as AM, are needed to match the accuracy of clinical annotations.

## DISCUSSION

Leveraging a large-scale IBD exome sequencing dataset, we showed, at single-variant and gene-based levels, that CF-risk variants in the *CFTR* gene confer a protective effect against IBD. This finding opens the possibility of developing selective, tissue-targeted CFTR modulators as a novel IBD therapeutic intervention. However, given that complete CFTR loss leads to CF, any therapeutic strategy must carefully balance efficacy with safety to avoid replicating the severe systemic consequences of CFTR deficiency.

Previously, Gabriel et al.[38] showed that heterozygous carriers of pathogenic *CFTR* variants are resistant to cholera toxin due to reduced intestinal fluid and chloride ion secretion in response to the toxin. We also know that the gut epithelial barrier plays a fundamental role in maintaining intestinal homeostasis and protecting against IBD. A major hallmark of IBD is the increased tight junction permeability and the thinning of the mucus layer, which allows bacterial translocation, thus triggering cascades of immune activation.[23,39–41] The protective effect of deltaF508 carriers could thus be achieved by regulating mucus hydration to maintain ideal viscosity for effectively trapping and clearing microbes and fluids, allowing for continuous renewal by fresh mucus. In addition, in the gut, *CFTR* is most highly expressed by BEST4+ enterocytes.[42,43] These cells have been implicated in maintaining gut homeostasis and are downregulated in active UC.[44] BEST4+ enterocytes of deltaF508 carriers would have

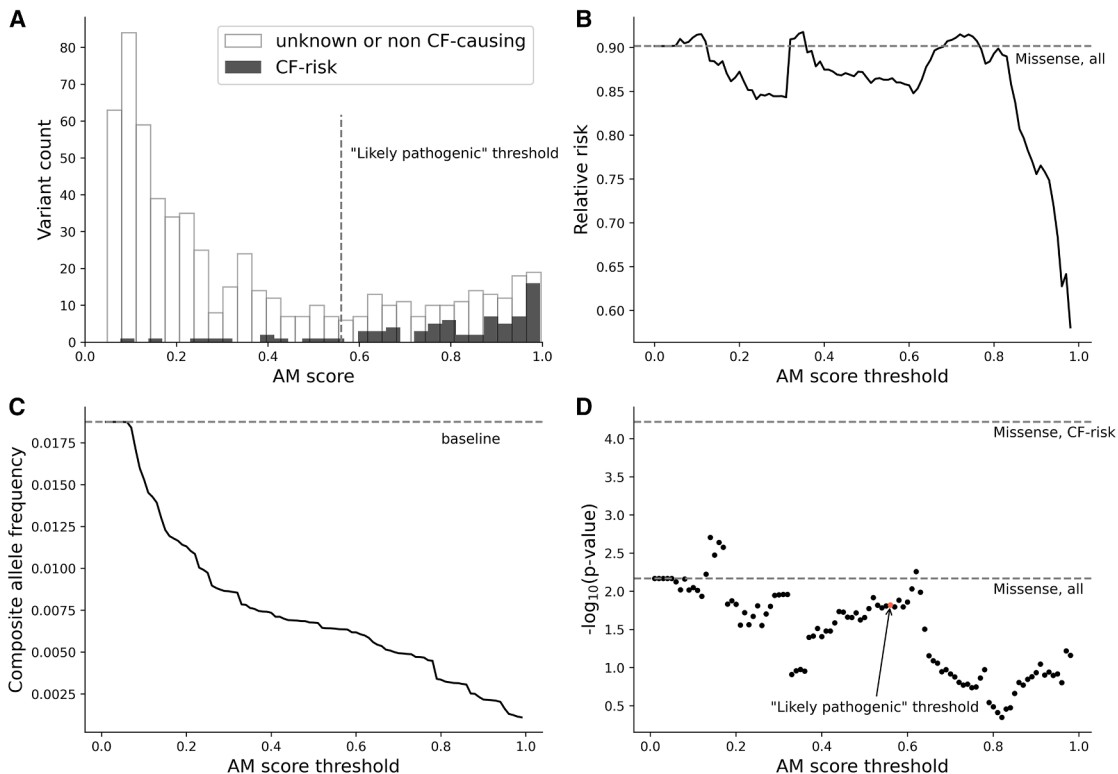

**Figure 3. Prioritizing *CFTR* variants in burden tests using AlphaMissense (AM). We restricted all tests to variants with MAF < 0.1%**

(A) AM scores for CF-risk variants and unknown or non-CF-causing variants.

(B) Relative risk for missense variants with AM score above the threshold (*x* axis). Relative risk was calculated as CAF in cases/CAF in controls.

(C) CAF for missense variants with scores above the AM threshold (*x* axis).

(D) Burden test significance using missense variants with AM scores above the threshold (*x* axis). The result using the AM "likely pathogenic" threshold (0.56) is marked in red.

lower *CFTR* expression and be less sensitive to stress or immune cues; e.g., via regulation of NF-kB signaling, cytokine secretion, response to oxidative stress, etc.[45] These hypoactive cells, under stress, might be buffered and operate in a low-reactivity zone that preserves their homeostatic function and survival.

In addition to its epithelial function, *CFTR* protein also regulates the gut microbiome.[14,15] A study showed that *CFTR* deletion in immune and epithelial cells influences the intestinal microbiota in mice.[46] It has been demonstrated that *CFTR* can serve as an epithelial receptor for *Salmonella* Typhi transluminal migration and that heterozygous deltaF508 mice translocated significantly fewer *Salmonella* Typhi into the GI submucosa than wild-type *CFTR* mice.[47] It is thus a plausible hypothesis that the protective effect of *CFTR* in IBD could be mediated through the microbiome.[32] Our finding suggests a previously under-appreciated role of CF-risk mutations in human health and disease. Follow-up experimental studies are warranted to investigate the exact cellular pathways CF-risk mutations disrupt to exert such a protective effect, as it may provide insights for an effective therapeutic intervention for IBD.

The Broad study ascertained control subjects from hospitals, while the Sanger study mostly ascertained controls from the community via the UK Biobank. Therefore, an over-representation of CF patients in the Broad EUR study can potentially create a spurious association between CF-risk variants and IBD. We controlled for this confounding factor by removing all predicted CF patients based on genetic data. To provide additional assurance, we estimated the impact of undetected CF patients in controls on our conclusion. While undetected CF patients can present in IBD patients as well, we applied this only to controls as a conservative measure. We found that an additional ∼200 CF-risk variant carriers in control would have had to be missed for the significance of the CF-risk variant burden test, with deltaF508 included, in the Broad EUR study to drop below the exome-wide significance threshold (Figure S2). Among the Broad controls, we identified 114 CF patients through genetic data. If 200 patients were missed, then this corresponds to a diagnostic sensitivity of 36% (114 diagnosed/[114 diagnosed +200 missing patients]) in Broad controls, much lower than the >99% sensitivity reported in the recent American College of Medical Genetics and Genomics (ACMG) statement,[48] which is very unlikely. Also, the CFTR2 database was curated using clinical and genetic information from 122,935 CF patients. Worldwide, 162,428 people are estimated to be living with CF.[49] With such a large sample size, the CFTR2 database is expected to have excellent coverage of CFTR causal variants. This is evident, as the pathogenic variant in CFTR2 has a MAF as low as 0.0005% (singleton in 122,935 CF patients), and the corresponding CAF, calculated using the MAF from the Broad controls, plateaued at around 1.2% (1.206% in 2022, 1.212% in 2023, and 1.291% in 2024). These are indications that the CFTR2 database is close to saturating the discovery of the more common CF-risk variants in the CF patient population. CF-risk variants missed in CFTR2, although possible, are extremely rare and are unlikely to lead to ∼200 undetected CF patients in controls to lower the diagnosis sensitivity to 36%. Therefore, our conclusion remains valid even though CFTR2 may not capture all CF-risk variants.

Furthermore, we replicated the protective effect of *CFTR* mutations against IBD in the EAS and AFR.AMR ancestry groups (Table 1). The South Asian ancestry group has very few subjects and was thus not included in this study. We performed burden tests combining all CF-risk variants, deltaF508 included, to maximize power considering the low frequency of deltaF508 and the much smaller sample sizes in these groups (STAR Methods). The CF-risk variant burden test replicated our finding at nominal significance for both the AFR.AMR group ($p = 0.023$, OR [95% CI] = 0.79 [0.65–0.97]; Table S6) and the EAS group ($p = 0.011$, OR [95% CI] = 0.16 [0.03–0.84]; Table S6). These results add an important note to the generalizability of our findings to other ancestries.

Burden tests are sensitive to the quality of variant annotations. While the clinical evidence-based *CFTR* variant annotations established the protective effect of CF-risk variants on IBD, simply including all predicted loss-of-function or missense variants in the burden test failed to reach exome-wide statistical significance. We evaluated AM, the best-in-class *in silico* variant pathogenicity predictor, and found that its effectiveness in variant prioritization for burden tests is hampered by insufficient accuracy. Further improvement of pathogenicity prediction models for missense variants is needed. Furthermore, a higher score threshold for selecting variants for burden tests does not guarantee improvement in statistical power, as it may sacrifice sensitivity for specificity, both of which are needed for a powerful burden test.

### Limitations of the study

The protective role of CF causal variants in IBD was identified through large-scale sequencing data. While this finding is robust, further studies are needed to elucidate the biological mechanisms underlying the protective effect. Our analyses in non-European populations remain limited by smaller sample sizes and the ongoing refinement of quality control (QC) and analytic strategies, particularly for the highly admixed AFR.AMR group. Expanding sample sizes in non-European populations and establishing robust analytic pipelines will be essential to understand CFTR's role in IBD across global populations.

### RESOURCE AVAILABILITY

#### Lead contact
Requests for further information and resources should be directed to and will be fulfilled by the lead contact, Hailiang Huang (hhuang@broadinstitute.org).

#### Materials availability
This study did not generate new, unique reagents.

#### Data and code availability
- All original code used in this manuscript is publicly available at https://doi.org/10.5281/zenodo.17355568, including scripts for genotype data QC, figure generation, and association analysis.
- Other computational tools used in this manuscript can be found in the key resources table.
- Availability for samples used in this study can be found in Table S1.

### CONSORTIA

Gonçalo Abecasis, Maria T. Abreu, Keyrun Adhikari, Waqqas Afif, Tariq Ahmad, Christopher Alexakis, Rebecca Ali, Anna Alkelai, Mohammed Allah-Ditta, Matthieu Allez, Mazer Ally, Eric Alm,

Silvia Alvarez, Ashwin N. Ananthakrishnan, Carl A. Anderson, Nikolas Apostolou, Richard Appleby, Muhammad Aqeel, Carmen Argmann, Ahmed Arslan, James Ashton, Joanna Atwater, Gil Atzmon, Marcus Auth, Ariane Ayer, Bijay Baburajan, Joshua Backman, Xiaodong Bai, Evelyn Baker, Michelle Baker-Moffatt, Suganthi Balasubramanian, Robert Baldassano, Antoine Baldassari, Tyara Banerjee, Paul Banim, Suying Bao, Aris Baras, Luis Barreiro, Arthur M. Barrie III, Nir Barzilai, Robert Battat, Ian Beales, Laurent Beaugerie, John Beckly, Ashley H. Beecham, Christina Beechert, Klaartje Bel Kok, Andrew Bell, Poonam Beniwal-Patel, Edmond-Jean Bernard, Charles N. Bernstein, James Berrill, Talat Bessissow, Roisin Bevan, Claire Bewshea, Meenakshi Bewtra, Madeleine Bezault, David G. Binion, Shrinivas Bishu, Alain Bitton, Pierre Blanc, Stuart Bloom, Michel Boivin, Bernd Bokemeyer, Krzysztof Borowski, Monica Bose, Simon Bouchard, Gabrielle Boucher, Mickael Bouin, Arnaud Bourreille, Boris Boutkov, Jonas Bovijn, Brendan Boyle, John Bradley, Steven R. Brant, Jose Bras, Carolyn Brechin, Biljana Brezina, Elaine Brinkworth, Johanne Brooks, Jessie Brown, Anthony Buisson, Kathy Burch, Ezra Burnstein, Deborah Butcher, Jeffrey Butterworth, Steven Buyske, Freddy Caldera, Adrian Campos, Franck Carbonnel, Christopher Cardinale, Erola Pairo Castineira, Melanie Caswell, Prathyusha Challa, Andrew Chan, Monica Chan, Eugene Chang, Katherine Chao, Annika Charlesworth, Colleen Chasteau, Lea Ann Chen, Lei Chen, Judy H. Cho, Sam Choi, Daniel Chung, Katie Clark, Isabelle Cleynen, Laura Cocking, Albert Cohen, Andrew (Andy) Cole, Joseph Collum, Nathan Constantine-Cooke, Rachel Cooney, Giovanni Coppola, David Corrigan, Lourdes Crane, Laura M. Cremona, Nicolas Croft, Raymond K. Cross, Fraser Cummings, David J. Cutler, Louise D'Aoust, Sushila Dalal, Allan Daly, Mark J. Daly, Oriana M. Damas, Amy Damask, Themistocles Dassopoulos, Lisa W. Datta, Nicholas O. Davidson, Albert Davies, Noor Dawany, Tina Day, Tanima De, Nanne K. De Boer, Kare De Felice, Juan de la Reveilla Negro, John DeCaestecker, Parakkal Deepak, Joanne Del Buono, Olivier Delaneau, Lee A. Denson, Aminda De Silva, Colette Deslandres, Érik Deslandres, Rofaida Desoki, Andrew Deubler, Anjan Dhar, Tanvi Dhere, Gerard Dijkstra, Angela Dobes, Sheila Dodge, Peter Dornbos, Hang Du, Shishir Dube, Jeffrey M. Dueker, Richard H. Duerr, Stacey Duffy, Ellyn Dunbar, Dharmaraj Durai, Ruth Eberhardt, Meltem Ece Kars, Evan Edelstein, Cathryn Edwards, David Ellinghaus, Eva Ellinghaus, Sarah Ennis, Gisu Eom, Laura Fachal, Martti Farkkila, William A. Faubion, Alison Fenney, Adolfo Ferrando, Manuel Ferreira, Eleanora A.M. Festen, Philip Fleshner, Stephen Foley, Caitlin Forsythe, Denis Franchimont, Andre Franke, Erin Fuller, Mathurin Fumery, Stacey B. Gabriel, Liron Ganel, Marco Gasparetto, Sheila Gaynor, Greta Gedgaudiene, Sahar Gelfman, Michel Georges, Benjamin Geraghty, Kyle Gettler, Arkopravo Ghosh, Maya Ghoussaini, Christopher Gillies, Arthur Gilly, Benjamin Glaser, Tessa Glazebrook, Sujit Gokhale, Lissette Gomez, John Gordon, Siegfried Görg, Alexander Gorovits, Manan Goyal, Philippe Goyette, Daniel B. Graham, Sarah Graham, Riley Grant, Anne Griffiths, Michael Grimes, Mathieu Groussin, Phillip Gu, Zhenhua Gu, Ju Guan, Rita Guerreiro, Kristy Guevara, Anton Gunasekera, Lauren Gurski, Mary Haas, Lukas Habegger, Hakon Hakonarson, Eija Hämäläinen, Anna Han, Laura Hancock, Jody Hankins, Mina Hanna, Talin Haritunians, Ailsa Hart, Samuel Hart, Alicia Hawes, Chris Hawkey, Gini Hay, Xavier Hebuterne, Caren Heller, Joseph Herman, Jaimee Hernandez, Rob Heuschkel, Mikko Hiltunen, George Hindy, Brian Hobbs, David Hobday, Jennifer Hollis, Patricia Hooper, Sami Hoque, Julie E. Horowitz, Hailiang Huang, David Hudesman, Jeffrey Hyams, Becky Icke, Peter Irving, Heba Iskandar, Yuval Itan, Vivek Iyer, Bana Jabri, Wisam Jafar, Chaim Jalas, Momodou W. Jallow, Mark Jarvis, Babur Javaid, Gbemisola Jenfa, Traci Jester, Kerrie Johns, Matthew (Matt) Johnson, Elyse Johnston, Emma Johnston, Laimas Virginijus Jonaitis, Kelsey Jones, Marcus B Jones, Eric Jorgenson, Tyler Joseph, Amit Joshi, Luke Jostins, Lijo Joy, Eugene Kalyuskin, Jochen Kammermeier, Akshay Kapoor, Manav Kapoor, Elizabeth W. Karlson, Jeffry Katz, Manreet Kaur, Maia Kayal, Judith Kelsen, Laura Kelylock, Nick Kennedy, Alexandra Kent, Michael Kessler, Hamed Khalili, Hossein Khiabanian, Nathalie Kingston, Julien Kirchgesner, Barbara S. Kirschner, Gediminas Kiudelis, Hannah Knight, Rita Kohen, Gauree Konijeti, Kimmo Kontula, Joshua Korzenik, Jukka T. Koskela, Jack Kosmicki, Konrad Koss, Ioannis Koumoutsos, Olga Krasheninina, Subra Kugathasan, Vijay Kumar, Juozas Kupcinskas, Satya Kurada, Alexander Lachmann, Peter Lakatos, Christopher A. Lamb, Nicola Lancaster, Rouel Lanche, Carol Landers, Karl Landheer, Jonathan Landy, Diane Langelier, Michael Lattari, Helena Lau, Matthias Laudes, Mark Lazarev, Lionel Le Bourhis, Michelle G. LeBlanc, Raymond Leduc, Ho-Su Lee, Jacinta Lee, James Lee, Kwang Lee, Charlie Lees, Neal LeLeiko, Michel Lemoyne, Kathryn (Katy) Leng, Chloé Lévesque, Rachel Levi, Adam P. Levine, Scott Levison, James D. Lewis, Stephen Lewis, Wendy Lewis, Andy Li, Dalin Li, Sheldon Lidofsky, Claire Liefferinckx, Jimmy Limdi, Nan Lin, Ruize Liu, Alan Lobo, Adam Locke, Britt-Sabina Loescher, Alexander Lopez, Luca A. Lotta, Edouard Louis, Christopher Macdonald, George Macfaul, David Mack, Vrushali Mahajan, Asif Mahmood, Zahid Mahmood, Sameer Malhotra, Kia Manoochehri, Adam J. Mansfield, John Mansfield, Dina Mansour, Jonathan Marchini, Anthony Marcketta, James Markowitz, Salvador Romero Martinez, Rebecca Matro, Jason Matthews, Evan K. Maxwell, Ernst Mayerhofer, Joelle Mbatchou, Jacob L. McCauley, Dermot P.B. McGovern, John McLaughlin, Simon McLaughlin, Gil Melmed, Emebet Mengesha, Miriam Merad, Virginie Mercier, Yin Miao, Raquel Milgrom, Lyndon Mitnaul, George Mitra, Paul Moayyedi, Christopher J. Moran, Gordon Moran, Deborah Morris, Arden Moscati, Craig Mowat, Ajay Muddu, Rafeeq Muhammed, Pooja Mule, Charles Murray, Mona Nafde, Priyanka Nakka, Stéphane Nancey, Rodney D. Newberry, William Newman, Darren Nix, Guillaume Noell, Chuka Nwokolo, Sirimon O'Charoen, Sean O'Keeffe, Susan O'Sullivan, Arabis Oglesby, David T. Okou, Bas Oldenburg, Tim Orchard, Katarzyna Orlicka, Harry Ostrer, Jacqueline Otto, Inès Ouchetati, John D. Overton, Nigel Ovington, Maria Sotiropoulos Padilla, Michelle Pagan, Sirish Palle, Joanne Palmer, William Palmer, Aarno Palotie, Jasmin Pandhal, Anita Pandit, Razvan Panea, Simon Panter, Benoît Panzini, Jean Paquette, Miles Parkes, Vinod Patel, Linda Patterson, Charles Paulding, Karine Paupert, Ashley Paynter, Joel Pekow, John Pekow, Laetitia Pele, Ruth Penn, Ann Perez-Beals, Inga Peter, Laurent Peyrin-Biroulet, Anne Phillips, Kath Phillis, Marieke J. Pierik, Vincent Plagnol, Richard Pollok, Cyriel Y. Ponsioen, Nikolas Pontikos, Jason Portnoy, Sam Powles, Mathilde Poyet, Manasi Pradhan, Meena Prasad,

Natalie Prescott, Cathryn Preston, Ann E. Pulver, Krishna Pawan Punuru, Maria A. Quintero, Shervin Rabizadeh, Laura E. Raffals, Monira Rahman, Souad Rahmouni, Tim Raine, Veera Rajagopal, Arvind Ramadas, John Ramage, Subramaniam Ramakrishnan, Nadia Rana, Venkatesh Ranganath, Ayesha Rasool, Paul Rastall, Jeffrey G. Reid, Elizabeth Renji, Joana Revez, Raymond Reynoso, Moen Riaz, Jennifer Rico-Varela, John D. Rioux, Marie-Ève Rivard, Johannie Rivera-Picart, Juan Rodriguez-Flores, Larizbeth Romero, Joel Rosh, Jonathan Ross, David Rubin, Kirk Russ, Päivi Saavalainen, Ksenija Sabic, Dariush Sadigh, Sumona Saha, George Salem, William Salerno, Bruce Sands, Michael Sargent, R. Balfour Sartor, Mudasar Sarwar, Jack Satsangi, John Saunders, Aleksejs Sazonovs, Eric Schadt, Ricardo Schiavo, Elena R. Schiff, Stefan Schreiber, L. Philip Schumm, Marc B. Schwartz, Randi Schwartz, Glyn Scott, Elizabeth A. Scoville, Shaji Sebastian, Anthony W. Segal, Philippe Seksik, Christian Selinger, Melanie Serrero, Sherif Shabana, Rakesh Shah, Mohammed Sharip, Dan Sharpstone, Rasha Shawky, Sophy Shedwell, Christopher Sheen, Shehzad Z. Sheikh, Richard Shenderey, Achuth Shenoy, Dhruv Shidhaye, Udi Shmueli, Richa Shukla, Alan Shuldiner, Yih-harn Siaw, Sunilbe Siceron, Carlo Sidore, Jon Silver, Mark S. Silverberg, Katherine Siminovitch, Alison Simmons, Rachel Simpkins, Claire L. Simpson, Salil Singh, Leena Sinha, Ganesh Sivaji, Kimberly Skead, Jurgita Skieceviciene, Alicia K. Smith, Katherine (Katie) Smith, Melissa Smith, Paul Smith, Phil J. Smith, Ryan Smith, Scott B. Snapper, Harry Sokol, Gina Solari, Matthew Solomonson, Hari Somineni, Kyuyoung Song, Thean Soon Chew, Olukayode Sosina, Michael Sprakes, Sanjay Sreeram, Eli Stahl, Helen Steed, Alan Steel, Joanne M. Stempak, Christine R. Stevens, Kathy Stirrups, Sreedhar Subramanian, Jae Soon Sul, Benjamin Sultan, Dylan Sun, Kathie Sun, Luanluan Sun, Mira Tang, Stephan R. Targan, Christopher Tastad, Byron Theron, Hartley Thomas, Iola Thomas, Julie Thompson, Timothy Thornton, Judith (Jude) Tidbury, Busiswa Titchmarsh, Philip Tombleson, Mark Tremelling, Xavier Treton, Karen Tricker, Dan Turner, Marcus Tutert, Gannie Tzoneva, Holm H. Uhlig, Ryan Ungaro, Andrea E. van der Meulen-de Jong, Kristel Van Steen, Deven Vani, Eric Vasiliauskas, Sai Lakshmi Vasireddy, Sailaja Vedantam, Suresh Venkateswaran, Ajay Verma, Séverine Vermeire, Marie-Pier Verpaelst, Sare Verstockt, Niek Verweij, Sophie Vieujean, Gareth Walker, Thomas Walters, Chen Wang, Chenggu Wang, Rujin Wang, Sophie Warden, Ben Warner, Kyoko Watanabe, Alastair Watson, Rinse K. Weersma, Zhi Wei, Frank Wendt, Emma Wesley, Katherine White, Gary Wild, Alan Wiles, Joy Wilkins, Cristen Willer, Horace Williams, Pauline Wils, David Wilson, Michael Wilson, Harland S. Winter, Sarah E. Wolf, Kuan-Han Wu, Ramnik J. Xavier, Shaohong Yang, Suk-Kyun Yang, Bin Ye, Byong Duk Ye, Justine Young, Mingrui Yu, Sean Yu, Kai Yuan, Samantha Zarate, Valentina Zavala, Aaron Zhang, Blair Zhang, Chuanyi Zhang, Jiayu Zhang, Jingning Zhang, Lance Zhang, Qian Zhang, Rui Zhang, Rufei Zhu, Mathias Zilbauer, David Ziring, Andrey Ziyatdinov, Yuxin Zou, Michael Zwick.

## ACKNOWLEDGMENTS

We thank all of the principal investigators, local staff from individual cohorts, and all of the patients who kindly donated samples used in the study for making this global collaboration and resource possible to advance IBD genetics research. We thank David Altshuler, Daniel Graham, and Harland Winter for helpful discussions. This research was funded in whole or in part by US National Institutes of Health grants U54HG003067 and 5UM1HG008895; Wellcome Trust grants 206194, 220540/Z/20/A, and 108413/A/15/D; and The Leona M. & Harry B. Helmsley Charitable Trust grant 2015PG-IBD001. We thank the Broad Institute Genomics Platform for genomic data generation efforts and the Stanley Center for Psychiatric Research at the Broad Institute for supporting control sample aggregation. We thank the Sanger Institute Scientific Operations teams and Human Genetics Informatics team for sample handling and data generation. This research was supported by the NIHR IBD BioResource and NIHR Biomedical Research Centres in Cambridge, Oxford, Imperial, UCH, and Newcastle. The NIHR IBD BioResource acknowledges co-funding by Crohn's and Colitis UK and The Leona M. and Harry B. Helmsley Charitable Trust. C.A.L. acknowledges research support from the NIHR Newcastle Biomedical Research Centre, Medical Research Council, The Leona M. and Harry B. Helmsley Charitable Trust, Crohn's & Colitis UK, the EU Innovative Medicines Initiative, the Wellcome Trust, Open Targets, the European Bioinformatics Institute (EMBL-EBI), Janssen, Takeda, Abbvie, AstraZeneca, Eli Lilly, Orion, Pfizer, Roche, Sanofi Aventis, UCB, Biogen, Genentech, Bristol Myers Squibb (BMS), GSK, and Merck Sharp and Dohme (MSD). H.H. acknowledges support from a Merkin Institute fellowship, the US National Institutes of Health grants K01DK114379 and R01DK129364, and the Stanley Center for Psychiatric Research. Individual studies contributing to this project acknowledge support from the US National Institutes of Health grants U01DK062431, U01DK062432, R01DK087694, K23DK117054, R01DK111843, P01DK094779, R01HG010140, U01HG009080, U01DK062420, P01DK046763, U01DK062413, P30DK043351, and R01DK104844. Participants in the INTERVAL randomized controlled trial (used as controls in the Sanger WGS dataset) were recruited with the active collaboration of NHS Blood and Transplant England (https://www.nhsbt.nhs.uk), which supported field work and other elements of the trial. DNA extraction and genotyping were co-funded by the National Institute for Health and Care Research, the NIHR BioResource (http://bioresource.nihr.ac.uk), and the NIHR Cambridge Biomedical Research Centre (BRC-1215-20014). The academic coordinating center for INTERVAL was supported by core funding from the NIHR Blood and Transplant Research Unit in Donor Health and Genomics (NIHR BTRU-2014-10024), NIHR BTRU in Donor Health and Behaviour (NIHR203337), the UK Medical Research Council (MR/L003120/1), the British Heart Foundation (SP/09/002, RG/13/13/30194, and RG/18/13/33946), and NIHR Cambridge BRC (BRC-1215-20014). A complete list of the investigators and contributors to the INTERVAL trial is provided in Di Angelantonio et al.[50] The academic coordinating center would like to thank blood donor center staff and blood donors for participating in the INTERVAL trial. This work was supported by Health Data Research UK, which is funded by the UK Medical Research Council, Engineering and Physical Sciences Research Council, the Economic and Social Research Council, the Department of Health and Social Care, the Chief Scientist Office of the Scottish Government Health and Social Care Directorates, the Health and Social Care Research and Development Division, the Public Health Agency, the British Heart Foundation, and Wellcome. The views expressed are those of the authors and not necessarily those of the NIHR, NHSBT, or the Department of Health and Social Care. The content is solely the responsibility of the authors and does not necessarily represent the official views of the National Institutes of Health.

## AUTHOR CONTRIBUTIONS

H.H., C.A.A., and M.J.D. designed and supervised the study. C.R.S., L.F., H.H., C.A.A., and M.J.D. were responsible for project management. M.Y., Q.Z., K.Y., A.S., and L.F. performed data analysis. M.Y., Q.Z., C.R.S., C.A.A., M.J.D., and H.H. wrote the manuscript. C.R.S. was responsible for sequencing technology development. C.A.L. was responsible for recruitment, clinical phenotyping, and leadership within contributing studies. C.A.L. revised the manuscript and contributed to the discussion of the molecular and cellular function of *CFTR*. All authors have reviewed and approved the manuscript.

## DECLARATION OF INTERESTS

C.A.L. acknowledges consultancy for MSD, Janssen, Takeda, and BMS; has received honoraria for development and/or delivery of education from Takeda, Ferring, Lilly, Janssen, Dr. Falk, and Nordic Pharma; and has received conference attendance support from Tillotts Pharma UK, Janssen, the British Society of Gastroenterology (BSG), the International Organization of IBD (IOIBD), and the European Crohn's & Colitis Organization (ECCO). C.A.A. has received research grants or consultancy/speaker fees from Genomics plc, BridgeBio, GSK, and AstraZeneca. M.J.D. is a founder of Maze Therapeutics.

## STAR★METHODS

Detailed methods are provided in the online version of this paper and include the following:

- KEY RESOURCES TABLE
- EXPERIMENTAL MODEL AND STUDY PARTICIPANT DETAILS
  - Study permissions and ethics
  - Participant details
  - Sample size
- METHOD DETAILS
  - Broad Institute data production
  - Sanger Institute data production
  - Clinical variant annotation
  - Variant annotation for molecular effect
  - Variant annotation using AlphaMissense
- QUANTIFICATION AND STATISTICAL ANALYSIS
  - Association analysis
  - Meta-analysis
  - Genetic association testing in an ancestrally admixed cohort
  - Local ancestry inference for admixed individuals
  - Power of replication studies

## SUPPLEMENTAL INFORMATION

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

## Article

**CellPress**

## STAR★METHODS

### KEY RESOURCES TABLE

| REAGENT or RESOURCE | SOURCE | IDENTIFIER |
|---|---|---|
| **Deposited data** | | |
| gnomAD | Karczewski et al.[51] | https://www.nature.com/articles/s41586-020-2308-7 |
| CFTR2 Variant List | [35] | https://cftr2.org/mutations_history |
| AlphaMissense pathogenicity scores | Cheng et al.[36] | https://zenodo.org/records/8208688 |
| 1000 Genomes (1KG) | 1000 Genomes Project Consortium[52] | https://www.nature.com/articles/nature15393 |
| Human Genome Diversity Project (HGDP) | Cann et al.[53] | https://www.science.org/ https://doi.org/10.1126/science.296.5566.261b |
| Sequence data used in this study | This paper | Genotypes and Phenotypes (dbGaP), accession codes: phs001642.v2.p1, phs002205.v1.p1, phs002018.v1.p1, phs001095.v1.p1, phs000572.v8.p4, phs000294.v1.p1, phs001489.v1.p1 https://www.donorhealth-btru.nihr.ac.uk https://www.ibdbioresource.nihr.ac.uk https://www.ukbiobank.ac.uk/use-our-data/apply-for-access/ https://duos.broadinstitute.org/datalibrary/ibd https://pmc.ncbi.nlm.nih.gov/articles/PMC6182437/ |
| **Software and algorithms** | | |
| Regenie | Mbatchou et al.[54] | https://github.com/rgcgithub/regenie |
| METAL | Willer et al.[55] | https://genome.sph.umich.edu/wiki/METAL |
| Hail | N/A | https://github.com/hail-is/hail |
| Scripts for quality control and analysis | This paper | https://doi.org/10.5281/zenodo.17355568 |
| **Other** | | |
| Members of the International Inflammatory Bowel Disease Genetics Consortium Sequencing Group | This paper | Table S7 |

## EXPERIMENTAL MODEL AND STUDY PARTICIPANT DETAILS

### Study permissions and ethics

All relevant ethical guidelines have been followed, and any necessary institutional review board (IRB) and/or ethics committee approvals have been obtained (Table S1). The Broad Institute component of this study was approved under Study Protocol 2013P002634 (the Broad Institute Study of Inflammatory Bowel Disease Genetics), and undergoes annual continuing review by the Mass General Brigham Human Research Committee IRB of Mass General Brigham (Mass General Brigham IRB). Approval was given on 27 January 2021 for this study. DNA samples sequenced at the Sanger Institute were ascertained under the following ethical approvals: 12/EE/0482, 12/YH/0172, 16/YH/0247, 09/H1204/30, 17/EE/0265, 16/WM/0152, 09/H0504/125, 15/EE/0286, 11/YH/0020, 09/H0717/4, REC 22/02, 03/5/012, 03/5/012, 2000/4/192, 05/Q1407/274, 05/Q0502/127, 08/H0802/147, LREC/2002/6/18, GREC/03/0273 and YREC/P12/0. All informed consent from participants has been obtained, and the appropriate institutional forms have been archived.

### Participant details

This study utilized a dataset comprising 340,028 samples collected across 81 independent studies (Table S1). All studies provided participant-reported gender information; however, our analysis was based on genetically inferred sex derived from the sequencing data, as this measure was available consistently across studies and aligned with our analytic framework. No additional demographic data (e.g., age, race, ethnicity, ancestry, or socioeconomic status) were requested from participants for this study. We assigned population ancestry by projecting all samples onto principal component (PC) axes generated from the 1000 Genomes Project (1KG) Phase 3 common variants, and classified their ancestry using a random forest method (method details).

### Sample size

Previous studies show that a large sample size is require to sufficiently power the IBD genetic discovery analysis.[21,56] Several recent studies of biobank scale suggest a sample size of over 100,000 individuals will be needed to properly power this study.[57,58] All available samples from collaborators throughout the IIBDGC were used and all replication samples available were included in downstream follow-up, as a result of this power estimate.

## METHOD DETAILS

### Broad Institute data production

#### Sequencing

The sequencing process included sample preparation (Illumina Nextera, Illumina TruSeq, and Kapa Hyperprep), hybrid capture (Illumina Rapid Capture Enrichment - 37 Mb target ["Nextera", Table 1], and Twist Custom Capture - 37 Mb target ["Twist"; Table 1]) and sequencing (Illumina HiSeq2000, Illumina HiSeq2500, Illumina HiSeq4000, Illumina HiSeqX, Illumina NovaSeq 6000, 76-base pair (bp) and 150-bp paired reads). Sequencing was performed at a median depth of 85% targeted bases at $>20\times$. Sequencing reads were mapped by BWA-MEM to the hg38 reference using the GATK 'functional equivalence' pipeline. The mapped reads were then marked for duplicates, and base quality scores were recalibrated. They were then converted to CRAMs using Picard 2.16.0-SNAPSHOT and GATK 4.0.11.0. The CRAMs were then further compressed using ref-blocking to generate gVCFs. These CRAMs and gVCFs were then used as inputs for joint calling. To perform joint calling, the single-sample gVCFs were hierarchically merged.

#### Quality control

Quality control (QC) analyses were conducted in Hail v.0.2.128 (Figure S1). Exome sequencing data (no whole-genome or blended genome exome) was first joint-called as part of the gnomAD database[51] (v4.1.0) and was later extracted out for QC and analysis. We first split multiallelic sites and coded genotypes with low genotype quality (GQ < 20) as missing. To exclude variant sites that fall outside of exome capture, we removed variants that are not annotated as frameshift, inframe deletion, inframe insertion, stop lost, stop gained, start lost, splice acceptor, splice donor, splice region, missense or synonymous. Sample-level QC. Samples that satisfy the following conditions were removed: (1) samples with an extremely large number of singletons ($\geq$500); (2) samples with mean GQ < 30; (3) samples with missingness rates >10%; (4) samples with outlying heterozygosity ($\pm$5 s.d. away from mean within the population); (5) samples with inconsistent genetically imputed sex and reported sex; and (6) duplicated samples, which were removed by identifying pairs of samples sharing identical genotypes (PI_HAT >0.95) and keeping the sample with higher mean GQ. Variant-level QC. Variants that satisfy following conditions were removed: (1) variants with missingness rate >5%; (2) variants with mean read depth (DP) < 10; (3) variants with mean genotype quality (GQ) < 30 (4) variants with >10% samples that were heterozygous and with an allelic balance ratio <0.3 or >0.7; and (5) variants that have known quality issues in both gnomAD v2 and v3 dataset (non-empty values in the "filter" column). Most of the parameters used in the QC procedures are chosen based on standards established in the literature, which have been shown to yield robust and reliable results.[21,59,60] In addition, we examined the variant-level QC metrics for all variant sites within the *CFTR* locus and found that the majority of variants have QC metrics well above the thresholds (Figure S3). In particular, the deltaF508 variant has an average read depth (DP) of 29.9, average genotype quality (GQ) of 41.3, and a call rate of 99.8%.

#### Ancestry assignment

After QC, we performed LD pruning on the synonymous variants removing variants with $R^2$ greater than 0.1 in a sliding window of 100,000 base pairs. This subset of low-correlation synonymous variants ($\sim$22,000) was then used to calculate the PCs for the 1KG Phase 3 subjects,[52] using the Singular Value Decomposition in Hail (v.0.2.128). A random forest model (number of trees = 100, Gini impurity as lost function, and unlimited tree depth) was trained using the top 10 PCs as features to predict 4 continental ancestry labels: European (CEU, TSI, FIN, GBR, IBS), African and American (YRI, LWK, GWD, MSL, ESN, ASW, ACB, MXL, PUR, CLM, PEL), East Asian (CHB, JPT, CHS, CDX, KHV), and South Asian (GIH, PJL, BEB, STU, ITU). 20% of the 1KG samples were held out for evaluation of the model's predictive performance, and a confusion matrix was generated. Overall, the model is able to predict ancestry labels well with a class error rate <1% in the testing set for all four classes. Finally, we projected all our subjects onto the PC space generated based on the 1KG Phase 3 subjects and used the top 10 PCs as input to the random forest model to assign each of our subjects to an ancestry bin. For subjects assigned to EUR, we only retained those with a prediction probability>80% as a conservative measure.

### Sanger Institute data production

#### Sequencing

Genome sequencing was performed at the Sanger Institute using the Illumina HiSeqX platform with a combination of PCR and PCR-free library preparation protocols. Sequencing was performed at a median depth of 18.6x. Exome sequencing of IBD cases was performed at the Sanger Institute using the Illumina NovaSeq 6000 and the Agilent SureSelect Human All Exon V5 capture set. Controls from the UK Biobank were sequenced separately as a part of the UKBB WES200K release using Illumina NovaSeq and the IDT xGen Exome Research Panel v1.0 capture set (including supplemental probes). 168,100 UKBB participants with EUR ancestry were selected for use as controls, excluding participants with recorded or self-reported CD, UC, unspecified noninfective

gastroenteritis or colitis, any other immune-mediated disorders, or a history of being prescribed any drugs used to treat IBD. Reads were mapped to hg38 reference using BWA-MEM 0.7.17. Variant calls were performed using DeepVariant and saved as per-sample gVCFs. These gVCFs are aggregated with GLnexus into joint-genotyped, multi-sample project-level VCFs (pVCFs). Variant calling was limited to Agilent extended target regions. Per-region VCF shards were imported into the Hail software and combined. This study considered variants located in intersection regions of Agilent and IDT exome captures +100bp buffer.

### Quality control

Exome sequencing: A combination of filters was used to identify low-quality variants and samples. Genotype calls with low genotype quality (GQ < 20) in rare variants (MAF <0.1%) were set as missing. A variant level QC was then applied by keeping variants that meet all the following criteria in both case (Sanger WES) and control (UKBB WES) samples: (1) mean GQ > 30; (2) mean read depth (DP) > 10; and (3) call rate >0.95. Samples satisfying any of the following conditions were removed: (1) low average GQ ($\leq$30); (2) low call rate ($\leq$0.9); (3) disagreement between genetically predicted and reported sex; (4) genetically identified duplicates (samples with higher call rate were retained); and (5) high rates of heterozygosity ($\pm$4 s.d. away from mean within each dataset). Genome sequencing: we applied variant quality score recalibration (VQSR) to calculate the variant quality score log-odds (VQSLOD) for each variant using GATK v4.4. Variants in the range of VQSLODs corresponding to the remaining 0.5% of the truth set were removed. Details on sample QC were available elsewhere.[21]

### Ancestry assignment

We selected a set of ~14,000 high-quality common variants that were shared between our subjects and 1KG Phase 3 subjects for ancestry assignment. Using this set of variants, we created four principal components from the 1KG Phase 3 subjects and projected our subjects to these components. We then used Random Forest to classify samples into continental genetic ancestry groups (EUR, AFR, SAS, EAS, admixed), with 1KG Phase 3 as the training dataset. We only retained the EUR samples for this study, as the number of cases for other ancestry groups was too small for robust association analysis.

## Clinical variant annotation

Cystic fibrosis variant annotation was downloaded from the Variant List History tab on CFTR2.org on April 23rd, 2024. Each variant was mapped to a variant ID in GRCh38 by its cDNA name.

## Variant annotation for molecular effect

Non-synonymous variants were classified using Ensembl Variant Effect Predictor[61] (VEP, v.95.0 in Broad institute and v.110.1 in Sanger institute)[61] as one of the following most severe consequences: "frameshift_variant", "stop_gained", "splice_acceptor_variant", "splice_donor_variant", "inframe_deletion", "inframe_insertion", "stop_lost", "start_lost", "missense_variant". Predicted loss-of-function variants include variants annotated as "frameshift_variant", "stop_gained", "splice_acceptor_variant", or "splice_donor_variant".

## Variant annotation using AlphaMissense

AlphaMissense (AM) predictions for all single amino acid substitutions in the human proteome data were downloaded and subsetted to 698 *CFTR* missense variants available in the Broad discovery dataset.

## QUANTIFICATION AND STATISTICAL ANALYSIS

### Association analysis

Association tests (both single variant and burden tests) were performed using a logistic mixed model implemented in REGENIE.[54] We chose this model because it accounts for sample size imbalances and sample relatedness, which was expected in this study. No analyses were performed in this study to determine whether the data meet assumptions of this method, but earlier work confirmed so on data of the same characteristics.[54] A set of high-confidence variants (>1% MAF, 99% call rate, LD-pruned with R2 = 0.9) was used for polygenic effect parameter estimation (Step 1). To control for effect size estimation bias caused by case-control imbalance, Firth correction (flags: "–firth –approx") was applied to association tests with $p$-value <0.05. To control for residual population structure, we calculated five PCs using a set of high-quality common SNPs, excluding regions with known long-range LD. In the Broad dataset, we additionally included a binary variable representing the two exome capture kits to control for the potential heterogeneity (Twist or Nextera). These variables were all included as covariates in both single-variant and burden tests, along with sex and the polygenic effect parameter calculated in Step 1. For single-variant analysis, association tests were only performed for variants with minor allele count $\geq$ 10. Burden tests were performed by collapsing pre-defined sets of variants (definition available in Tables 2 and 3; Figure 3) into single combined "mask" genotypes, which are computed as the sum of unweighted allele dosages across all variants in the set. All $p$-values from the association tests were computed based on a Wald test, where the test statistics under the null hypothesis follow a chi-square distribution with 1° of freedom. Details of the analyses can be found in Results.

### Meta-analysis

We used an inverse variance-weighted fixed-effect model implemented in METAL[55] to meta-analyze the association statistics, including single-variant and the burden test, across different studies. While not tested in this study, earlier work showed that the inverse variance-weighted fixed-effect model is appropriate for IBD genetic analyses.[21] Details of the analyses can be found in Results.

**Cell Genomics**
**Article**

### Genetic association testing in an ancestrally admixed cohort

Individuals in the AFR.AMR cohort have varying admixture proportions of African and American ancestry, forming a continuum in the PC space (Figure 1). To ensure the robustness of variant testing in admixed populations, a common approach is to perform local ancestry inference (LAI) and include in the regression model the terms representing the number of risk alleles coming from each ancestry.[62] This allows the model to estimate ancestry-specific effect sizes and control for allele frequency differences that are driven by ancestry. However, this approach is not feasible here since most CF-risk variants are too rare for phasing results to be reliable. As an alternate plan, we performed the CF-risk variant burden test for the AFR.AMR cohort by including the LAI parameters in the *CFTR* locus as covariates. We reason that, though not a perfect approach, it mitigates the potential confounding effect that local population structure has on case and control allele frequency. Encouragingly, we also observe that there is minimal difference in local population structure in the *CFTR* locus between CF-risk variant carriers and non-carriers (Figure S4), suggesting that any association between CF-risk variants and disease status is very unlikely to be driven by the population structure.

### Local ancestry inference for admixed individuals

For samples assigned to the African and American (AFR.AMR) ancestry group, we inferred local genomic ancestry at the *CFTR* locus. The sample genotype data were first phased using SHAPEIT5,[63] with the harmonized 1KG phase 3 and Human Genome Diversity Project[53] (HGDP) dataset as the reference panel.[64] Subsequently, we used FLARE[65] to perform local ancestry inference using African, European, and Native American populations from the HGDP + 1KG dataset as the ancestry reference panel.

### Power of replication studies

In the genetic association analysis, the test statistic for an association between an allele and a trait follows a noncentral chi-squared distribution. The Non-Centrality Parameter (NCP) is

$$\lambda = \frac{N \cdot 2f(1-f) \cdot \beta^2}{\sigma^2}$$

where $N$ is the effective sample size, calculated as $4/(1/N\_cases + 1/N\_controls)$, $f$ is the allele frequency, $\beta$ is the effect size estimate of the allele, and $\sigma$ is the standard deviation of the trait. Assuming that the effect size estimate is the same in the discovery and replication studies, the NCP for the replication study is proportional to the one in the discovery study:

$$\lambda_1 = \lambda_0 \frac{N_1 \cdot f_1(1-f_1)}{N_0 \cdot f_0(1-f_0)}$$

where $\lambda_0, f_0, N_0$ are the NCP, allele frequency, and sample size for the discovery study, and $\lambda_1, f_1, N_1$ are the NCP, allele frequency, and sample size for the replication study. The power of the replication study is then:

$$\text{Power} = P\left(\chi_1^2(\lambda_1) > \chi_{\alpha,1}^2\right) = F_{\chi_1^2(\lambda_1)}\left(\chi_{\alpha,1}^2\right)$$

where F is the survival function of the non-central chi-square distribution (degree of freedom = 1) and $\alpha$ (=0.05) is the significance threshold.

