## [Document S2. Transparent peer review records for Yu et al. · Cell Genomics]

Summary

Initial submission: Received : 12/18/2024

Scientific editor: Laura Zahn

First round of review: Number of reviewers: 2
Revision invited : 1/28/2025
Revision received : 5/7/2025

Second round of review: Number of reviewers: 2
Accepted : 10/26/2025

Data freely available: Yes

Code freely available: Yes

This transparent peer review record is not systematically proofread, type-set, or edited. Special characters, formatting, and equations may fail to render properly. Standard procedural text within the editor's letters has been deleted for the sake of brevity, but all official correspondence specific to the manuscript has been preserved.

Referees' reports, first round of review:

Reviewer #1: Overall, the authors identified cystic fibrosis risk variants confer protection against inflammatory bowel disease.

I have some specific questions and suggestions as follows.

1. Quality control criteria explanation: The manuscript should provide a clearer explanation of the quality control criteria used for variant filtering, such as the thresholds for mean GQ > 30, mean read depth (DP) > 10, and call rate > 0.9. It would be beneficial to include more details about why these specific thresholds were chosen and how they impact the study's results.

2. Sample selection justification: The authors should justify why only EUR samples were retained for the study. Elaborating on the reasons for excluding other ancestry groups, particularly in terms of the number of cases and the robustness of association analysis, would strengthen the manuscript.

3. PCA details: The PCA used for ancestry assignment should be described in more detail. It would be helpful to see a supplementary figure or table showing the distribution of the samples along the principal components and the clear separation between different ancestry groups.

4. Random forest methodology: The manuscript should provide more information on the Random Forest model used for classifying samples into genetic ancestry groups. This includes details on model parameters, feature selection, and validation methods to ensure the robustness of the classification.

5. Supplementary figures: The supplementary figures (1 and 2) should be better integrated into the main text. Reference to these figures should be made in the appropriate sections of the manuscript to help readers understand the context and results more clearly.

Reviewer #2: This manuscript presents a novel and significant contribution to understanding the genetic relationship between cystic fibrosis (CF) and inflammatory bowel disease (IBD). The study leverages large-scale exome sequencing datasets to suggest that CFTR gene mutations, typically associated with cystic fibrosis, may confer protection against IBD. The study's findings are robust, relying on a substantial amount of data (38,558 IBD cases and 66,945 controls in the discovery stage), and the replication in independent datasets adds credibility to the results. The approach is scientifically sound and has potential clinical implications. However, I still have some questions.

1. Theoretical Framework: While the authors have outlined a plausible biological mechanism for how CFTR mutations might protect against IBD, this could be explored in more depth. More detailed discussion of how CFTR mutations affect immune responses or epithelial cell function in the gut could further strengthen the study's implications.

2. Experimental Validation: Although the study provides strong statistical evidence, experimental validation using cell culture or animal models would further corroborate the findings. This could address potential concerns regarding the biological mechanisms driving the observed associations.

3. Generalization to Other Populations: The study predominantly uses European ancestry samples. The authors should consider discussing how these findings may or may not generalize to populations with different genetic backgrounds, such as African or Asian populations.

4. Burden Test Results: The burden test results for non-CF-risk variants were not as compelling as those for CF-risk variants. A more thorough explanation of why non-CF-risk variants did not show significant associations with IBD would add clarity. Additionally, the authors could explore whether these variants have an indirect role in other diseases.

5. Variant Selection: The manuscript mentions that CF-risk variants were selected based on the CFTR2 database annotations. However, there could be additional variants that were missed due to limitations in the CFTR2 database or the annotation process. How confident are the authors that all relevant CF-risk variants were included?

6. Control for Ascertainment Bias: The authors addressed potential biases from the study design (e.g., over-representation of CF patients in the Broad dataset). It would be helpful to include a more detailed sensitivity analysis or simulation to assess how the control population might still impact the findings, especially since the exclusion of CF patients was based on genetic data approximations.

7. Statistical Power in Replication Cohorts: The Sanger WGS dataset, being smaller, showed

weaker associations. Could the authors perform additional power analysis to further validate the robustness of these findings, especially in smaller datasets?

Authors' response to the first round of review

We thank all reviewers for their insightful comments and constructive suggestions. We have carefully considered the points made by the reviewers and revised the manuscript accordingly. We have provided the following point-by-point response to specific reviewer concerns. Our response is colored in light green for ease of review. Changes to the text are also highlighted in light green in the updated manuscript.

Reviewer #1: Overall, the authors identified cystic fibrosis risk variants confer protection against inflammatory bowel disease.

I have some specific questions and suggestions as follows.

1. Quality control criteria explanation: The manuscript should provide a clearer explanation of the quality control criteria used for variant filtering, such as the thresholds for mean GQ > 30, mean read depth (DP) > 10, and call rate > 0.9. It would be beneficial to include more details about why these specific thresholds were chosen and how they impact the study's results.

We adapted the parameters used in our QC procedures from the standards established in the previous high-quality studies and literature, which have been demonstrated to yield robust and reliable results. In response to the concerns, we have evaluated the distribution of variant-level metrics with respect to our thresholds and found these thresholds are effective in removing outliers and retaining high-quality sequencing reads. We have expanded the discussion and included citations to relevant studies where these parameters were applied (Methods, lines 482-493, Supplementary Figure 3).

2. Sample selection justification: The authors should justify why only EUR samples were retained for the study. Elaborating on the reasons for excluding other ancestry groups, particularly in terms of the number of cases and the robustness of association analysis, would strengthen the manuscript.

We appreciate the reviewer's suggestion. We initially excluded non-EUR groups because of their much smaller sample sizes and the fact that deltaF508 is much more commonly seen in European populations. However, we found that the non-European groups, especially the African and American admixed population (AFR.AMR), actually have a reasonable power to replicate the CFTR association using the burden test. Therefore, we have added a discussion of non-European analysis in the revised manuscript (lines 328-337). Encouragingly, the burden test in both AFR.AMR and EAS ancestries, using CF-risk variants, replicated the association between CFTR and IBD at nominal significance, adding an important note to the generalizability of this finding to other ancestral populations.

3. PCA details: The PCA used for ancestry assignment should be described in more detail. It would be helpful to see a supplementary figure or table showing the distribution of the samples along the principal components and the clear separation between different ancestry groups. We have moved the PC figure to the main text (Figure 1) and added histograms to better visualize the distribution of samples across different ancestry groups along the PCs. We have also added more details on the calculation of PC used for ancestry assignment and the variant selection that precedes it in Methods (lines 495-499, 506-510).

4. Random forest methodology: The manuscript should provide more information on the Random Forest model used for classifying samples into genetic ancestry groups. This includes details on model parameters, feature selection, and validation methods to ensure the robustness of the classification.

We thank the reviewer for this suggestion. We have included more details on the Random Forest model as suggested (lines 499-506).

5. Supplementary figures: The supplementary figures (1 and 2) should be better integrated into the main text. Reference to these figures should be made in the appropriate sections of the manuscript to help readers understand the context and results more clearly.

Those figures are now more appropriately cited to be better integrated into the main text. PC figure, which was originally supplementary figure 2, is now a main figure. It has also been revised to include the PCs for samples from other ancestry groups in our dataset.

Reviewer #2: This manuscript presents a novel and significant contribution to understanding the genetic relationship between cystic fibrosis (CF) and inflammatory bowel disease (IBD). The study leverages large-scale exome sequencing datasets to suggest that CFTR gene mutations, typically associated with cystic fibrosis, may confer protection against IBD. The study's findings are robust, relying on a substantial amount of data (38,558 IBD cases and 66,945 controls in the discovery stage), and the replication in independent datasets adds credibility to the results. The approach is scientifically sound and has potential clinical implications. However, I still have some questions.

1. Theoretical Framework: While the authors have outlined a plausible biological mechanism for how CFTR mutations might protect against IBD, this could be explored in more depth. More detailed discussion of how CFTR mutations affect immune responses or epithelial cell function in the gut could further strengthen the study's implications.

We thank the reviewer for this suggestion. We have included a thorough discussion of the potential biological mechanisms for the protective effects of CFTR mutations against IBD (lines 246-291).

2. Experimental Validation: Although the study provides strong statistical evidence, experimental validation using cell culture or animal models would further corroborate the findings. This could address potential concerns regarding the biological mechanisms driving the observed associations.

While we agree with the reviewer that experimental investigation of our findings is very valuable, we note that there have already been extensive experimental studies on the molecular and cellular functions of CFTR, as well as the impact of deltaF508 on human physiology. These studies provide hypotheses that can be tested through experiments using cell culture or animal models to reveal the biological mechanism underlying the protective effect of defective CFTR against IBD. We have summarized these studies comprehensively (lines 246-291). We argue, respectfully, that performing experiments to test these hypotheses would be beyond the scope of this study.

3. Generalization to Other Populations: The study predominantly uses European ancestry samples. The authors should consider discussing how these findings may or may not generalize to populations with different genetic backgrounds, such as African or Asian populations.

We appreciate the reviewer's suggestion. We initially excluded non-EUR groups because of their much smaller sample sizes and the fact that deltaF508 is much rarer in non-European populations. However, we found that the non-European groups, especially the African and American admixed population (AFR.AMR), actually have a reasonable power to replicate the CFTR association using the burden test. Therefore, we have included an additional discussion of non-European analysis in the revised manuscript (lines 328-337). Encouragingly, the CF-risk

variant burden test in both AFR.AMR and EAS cohorts replicated at nominal significance the CFTR-IBD association. This adds an important note to the generalizability of this finding to other ancestral populations.

4. Burden Test Results: The burden test results for non-CF-risk variants were not as compelling as those for CF-risk variants. A more thorough explanation of why non-CF-risk variants did not show significant associations with IBD would add clarity. Additionally, the authors could explore whether these variants have an indirect role in other diseases.

We apologize for the confusion. We have clarified the rationale behind including the non-CF-risk variant burden test and its lack of significant association with IBD. CF-risk variants are defined as mutations that impact CFTR protein function, leading to CF. In contrast, non-CF-risk variants are CFTR mutations not associated with CF and are presumed to have no functional effect. The non-CF-risk variant burden test serves as a negative control, reinforcing our hypothesis that only mutations affecting CFTR protein function are protective against IBD. We have updated the manuscript to avoid this confusion (line 141-145, 175-178).

5. Variant Selection: The manuscript mentions that CF-risk variants were selected based on the CFTR2 database annotations. However, there could be additional variants that were missed due to limitations in the CFTR2 database or the annotation process. How confident are the authors that all relevant CF-risk variants were included?

The reviewer raised a very important question. We have given these thorough thoughts and have added additional analysis in response.

First, the CFTR2 database was curated using clinical and genetic information from 122,935 CF patients. Worldwide, 162,428 people are estimated to be living with CF (Guo et al., *J Cyst Fibros* 21, 456–462, 2022). With a sample size of ~75% of the living patient population, CFTR2 database is expected to have excellent coverage of CFTR causal variants. For instance, as a naive estimate, the probability for a triplet (a variant carried by 3 patients worldwide, with the estimated frequency of $9E-6$ in patients) to be missed in CFTR2 is only 1.5% (0.25^3) (admittedly, this is a gross estimate as it assumes perfect sequencing technology to capture the variants and etc).

In addition, the minor allele frequency of the rarest “CF-causing” variant captured in CFTR2 was as low as 0.0005% in its patient population (singleton in 122,935 CF patients). The number of “CF-causing” or “Varying clinical consequences” variants in CFTR2, even at low frequency (0.01%), saw very little increase from 2022 to 2024 (table below). The corresponding CAF in the Broad EUR study has also plateaued at around 1.2% at the same time. These are indications that CFTR2 is close to saturating the discovery of the more common CF-risk variants in the CF patient population. While it is possible that there are additional variants not included in CFTR2 that also confer risk to CF, those variants are extremely rare in the population. Therefore, failing to include these variants should have a limited impact on establishing the protective role of CFTR on IBD.

2022 2023 2024

“CF-causing” or “Varying clinical consequences”

(AF in CF population > 0.01%) 210 210 215

composite allele frequency 0.01206 0.01212 0.01291

Lastly, because our hypothesis is that mutations causal to CF are protective against IBD, missing CF-risk variants reduces our CAF and thus our power to detect this hypothesized association. But it won't inflate false-positive rates and thus will not affect our conclusion (except for comment #6, see response below).

We have added these discussions to lines 314-326.

6. Control for Ascertainment Bias: The authors addressed potential biases from the study design

(e.g., over-representation of CF patients in the Broad dataset). It would be helpful to include a more detailed sensitivity analysis or simulation to assess how the control population might still impact the findings, especially since the exclusion of CF patients was based on genetic data approximations.

We thank the reviewer for this important suggestion. As there have been no known and credible causes of CF other than inheriting two copies of the defective CFTR gene, the undetected CF patients may only be due to the incomplete list of CF causal variants from CFTR2. We therefore performed a simulation where a number of controls are “undetected CF patients” that should have been excluded. Results from this analysis showed that our conclusion is robust even with an incomplete list of CF causal variants (lines 305-314).

7. Statistical Power in Replication Cohorts: The Sanger WGS dataset, being smaller, showed weaker associations. Could the authors perform additional power analysis to further validate the robustness of these findings, especially in smaller datasets?

We have calculated and reported the statistical power for the replication analysis as suggested. Indeed, Sanger WGS has a smaller power to replicate the burden test finding at the nominal significance compared with Sanger WES (41% versus 99%), as the reviewer expected. We have also added additional subjects from non-European populations as replications, which provide additional assurance in the robustness of the conclusion, thanks to the reviewer’s other comment. We have added the results to the manuscript (lines 103 and 166-167).

Referees’ report, second round of review:

Reviewer #3: Accept

Reviewer #4: In their study, Yu et al. investigate the potential protective impact of CF-related CFTR variants on IBD through whole exome and genome sequencing data in different cohorts. The comments from the previous reviewers have partly been addressed, but there are still details, especially methodological, missing in the manuscript.

The burden tests are not introduced at all in the introduction, and the authors did not justify their use in the present study. Similarly, no detail is given in the methods on how the burden tests were performed: what was the MAF range considered, which type of burden, what statistical test was done? If I understand correctly, variants with a MAF lower than 0.1% were included, why such a low threshold?

For me, it is really not clear which variants were included in the single-point analyses, and which one in the burden tests. For the single-variant associations, how many tests were done and was correction for multiple testing applied?

The authors integrated the comment on extension to non-European ancestry groups, but corresponding analyses only come quite late in the discussion and should be added in the results. Looking at the PCA, the AFR.AMR group is heterogeneous, and I guess it’s because AFR and AMR have been combined. Why did the authors not consider the two groups separately? Also, what were the results on deltaF508 alone, and on burden tests without this variation? Analyses should be consistent with the discovery one.

Additional methodological points:

- Why was there no meta-analysis on the non-CF-risk variants?**
- Why only synonymous variants were considered on the PCA? There should not be any filter on consequence**
- Please provide PCA plots with the repartition of cases and controls, at least for the discovery analyses to ensure there is no bias**
- l. 215: of the 73% variants, are they non-risk or not present in the database? If the latter, this**

does not inform on the pathogenicity of the variants. Comparison of AM scores and CFTR2 status should restrict to the variants present in both. Alternatively, enrichment analyses could be performed to compare the two.

The results are not put into context. For example, what was known about the genetics of IBD? Are the effect sizes of CFTR variants on IBD similar to what has been previously described in IBD studies, i.e. how much these variants play in genetic IBD risk?

Overall, some rewriting and ordering of the manuscript is needed:

- It is really hard to follow the manuscript between IBD, CD, UD and in consistent (for example, no mention of UD in Table 3). What is the main phenotype of interest?
- I.175-177: the justification should come earlier in the manuscript, it is weird to give it after the results

- I. 200-209: this paragraph should come earlier as well

- There is no need to specific point to "Discussion" in different parts of the paper

- L. 350-353: The conclusion of the discussion does not match the goals of the paper

- The discussion is quite long and some things could be moved to the introduction

Additional precisions/changes:

- L. 122: remain significant at which threshold?

- In Table 2, please give OR in addition of its CI

- I. 312-314: this is not clear

- Is the pipeline for association test between Broad and Sanger the same? If so, please simplify in the methods, if not, please explain why.

- I.179: in this study, there is no assessment of the function of CFTR variants considered

- There is no reference for everything related to BEST4+

- I don't think that the CFTR2 database really contains 75% of living patients as probably the database gathers patient information from previous decades.

- On the different filtering strategies (Table 3), it would be interesting to see how the variant sets compare to each other and to CFTR2 (as it was done for AM in paragraph I. 211 for example)

Authors' response to the second round of review

an additional and independent piece of evidence

supporting the protective role of CFTR mutations against IBD, complementing the single-variant association analysis of delF508. We have now described the burden test in the Introduction and added more details in Methods and Results. We selected the MAF threshold of 0.1% based on empirical evidence from prior studies that low-frequency coding variants are more likely to have strong effects on traits (likely due to selection) and less likely to be affected by the linkage disequilibrium, leading to improved statistical power and calibrated false-positive rate in the burden test¹⁻³.

2. For me, it is really not clear which variants were included in the single-point analyses, and which one in the burden tests. For the single-variant associations, how many tests were done and was correction for multiple testing applied?

We apologize for the confusion. In Supplementary Table 3 (ST3), we listed all variants in the CFTR locus that were captured in the Broad dataset. For burden tests, depending on the choice of variant sets (e.g., CF-risk, predicted loss-of-function, etc), we selected subsets of variants among variants in ST3 using criteria now clearly described in the manuscript.

Because we discovered the delF508 association through an exome-wide study, we applied a conservative exome-wide significance threshold of $p=1 \times 10^{-6}$.

For single variant associations, we only performed tests for variants with minor allele count (MAC) ≥ 10 (case and control combined). Variants have a valid P-value in ST3 if tested, and "-" if not. Besides delF508, 123 single variants were tested. We only used these 123 variants for visualization and the sign test in aggregation (binomial test). Therefore, they do not need to be corrected for multiple testing.

We used the nominal significance threshold ($P=0.05$, no multiple testing correction) for the burden test, because it was only performed on CFTR excluding delF508.

We have added the significance thresholds to the main text to avoid confusion.

3. The authors integrated the comment on extension to non-European ancestry groups, but corresponding analyses only come quite late in the discussion and should be added in the results. Looking at the PCA, the AFR.AMR group is heterogeneous, and I guess it's because AFR and AMR have been combined. Why did the authors not consider the two groups separately? Also, what were the results on deltaF508 alone, and on burden tests without this variation? Analyses should be consistent with the discovery one.

We thank the reviewer for these thoughtful questions. We analyzed AFR and AMR groups together because they represent a highly admixed and continuous group, as the PC plot showed. Furthermore, the sample size for AMR.AFR is relatively small. If we split them into two groups, we will lose subjects that cannot be unambiguously assigned to either, further reducing the sample size. As such, we respectfully retain our strategy for treating AMR.AFR as one group in this study.

Considering the much smaller sample sizes for the AFR/AMR and EAS groups, and the much lower allele frequency of delF508 outside of European populations, we did not perform the single-variant test for delF508 in AFR/AMR nor EAS groups, as the statistical power will not be sufficient. Instead, we included delF508 in the CFTR burden test, which is a strategy different from that of the European population, but it is necessary considering the sample size and allele frequency.

We presented the analysis in non-European populations in the Discussion rather than integrating them in the Results because we consider these findings secondary evidence. In contrast to the analyses in European populations—supported by large sample sizes and a well-validated QC and analytic pipeline—the analyses in non-European populations are limited by smaller sample sizes and the ongoing evaluation of optimal QC and analytic strategies, particularly for the highly admixed AFR/AMR group. For example, questions remain regarding the sensitivity and accuracy of local ancestry inference and the effectiveness of controlling for local ancestry in the regression models. While we are confident in the quality of our current analyses and findings, we believe a systematic investigation of these populations—addressing these methodological questions through simulations and including other traits as positive and negative controls—should be undertaken. Such work is beyond the scope of the present study and is better suited for a separate paper. For this reason, we chose to present the non-European results in the Discussion rather than in the Results, treating them as secondary evidence.

Additional methodological points:

We thank the reviewer for carefully reading the manuscript and providing thoughtful questions regarding the methodologies.

4. Why was there no meta-analysis on the non-CF-risk variants?

We have now included the meta-analysis on the non-CF-risk variants in Table 3, which was not significant as expected (p -value = 0.06).

5. Why only synonymous variants were considered on the PCA? There should not be any filter on consequence

We only included synonymous variants in computing the PCs for ancestry group assignment because nonsynonymous variants are more likely under selection for their impact on a protein's molecular function, and can capture non-population factors.

Restricting to synonymous variants is a safe choice and is not expected to reduce PC's

ability to capture the population structure.

6. Please provide PCA plots with the repartition of cases and controls, at least for the discovery analyses to ensure there is no bias

We have replaced the original PCA plot with a new one where cases and controls are separately shown, making it easier to inspect any bias visually.

7. l. 215: of the 73% variants, are they non-risk or not present in the database? If the latter, this does not inform on the pathogenicity of the variants. Comparison of AM scores and CFTR2 status should restrict to the variants present in both. Alternatively, enrichment analyses could be performed to compare the two.

While we agree that absence from CFTR2 does not equate to being classified as non-risk, such absence strongly suggests (though not conclusively) that a variant is non-risk. This is because CFTR2 provides excellent coverage of causal variants, particularly those with large and clinically meaningful effects.

Our observation confirmed this: 26% variants not in CFTR2 have $AM \geq 0.56$. This proportion is statistically indistinguishable from the proportion for variants classified as "Non CF-causing"/"Unknown significance" (25%, p -value=0.64), and much lower than the rates for "CF-causing"/"Varying clinical consequences" variants (84%, p -value < $2E-16$).

We have separated the 73% variants by their CFTR2 annotations and revised our manuscript accordingly for clarity.

8. The results are not put into context. For example, what was known about the genetics of IBD? Are the effect sizes of CFTR variants on IBD similar to what has been previously described in IBD studies, i.e. how much these variants play in genetic IBD risk?

We have included more background on IBD genetics and a discussion of known mutations that also confer protection against IBD in the Introduction section, which we believe will provide more context for the findings that were introduced later.

9. Overall, some rewriting and ordering of the manuscript is needed:

- It is really hard to follow the manuscript between IBD, CD, UD and in consistent (for example, no mention of UD in Table 3). What is the main phenotype of interest? We apologize for the confusion. The main phenotype of interest is IBD. CD and UC are two subtypes of IBD. We tested CD, UC and IBD when possible for *delF508*, to provide the full evidence to support our conclusion. The only exception is in the Sanger WGS dataset: we only tested the CD association as all cases were CD patients, which is why the phenotype was originally labeled "CD/IBD." We have now added a note in the Table legends to avoid ambiguity ("Sanger WGS only included CD patients"). For the burden test, we only presented findings for IBD in the main table (Table 3) for simplicity, as we tested various functional categories and do not want to clutter the table. The burden test results for CD and UC are separately provided in Supplementary Table 4. We have rewritten the manuscript for clarity.

10. - l.175-177: the justification should come earlier in the manuscript, it is weird to give it after the results

We have restructured the paragraph accordingly to make it more coherent.

11. - l. 200-209: this paragraph should come earlier as well

We have moved this paragraph to the start of the subsection.

- There is no need to specific point to "Discussion" in different parts of the paper

We have removed the reference to the Discussion section throughout the paper.

12. - L. 350-353: The conclusion of the discussion does not match the goals of the paper

We have removed this paragraph so that it doesn't distract the reader from the main conclusion.

13. - *The discussion is quite long and some things could be moved to the introduction*
We have restructured the Introduction and Discussion sections so that it is more readable and balanced.

Additional precisions/changes:

14. - L. 122: *remain significant at which threshold?*

*We have clarified in the manuscript that the association between *delf508* and IBD remains exome-wide significant after meta-analysis.*

15. - *In Table 2, please give OR in addition of its CI*

We have added ORs with 95% CI to all the tables with association test statistics.

16. - *I. 312-314: this is not clear*

We have rewritten this sentence.

17. - *Is the pipeline for association test between Broad and Sanger the same? If so, please simplify in the methods, if not, please explain why.*

The association testing pipelines for the Broad and Sanger datasets are identical, except that in the Broad dataset we included an additional regression covariate indicating sequencing platform (Twist or Nextera) to account for platform-related heterogeneity. We have clarified and simplified this description in the Methods section.

18. - *I.179: in this study, there is no assessment of the function of CFTR variants considered*

While indeed there was no assessment of their molecular function, these variants were evaluated for their CF-related physiological functions. We have reworded for accuracy.

19. - *There is no reference for everything related to BEST4+*

We have included references regarding the discussion related to BEST4+.

20. - *I don't think that the CFTR2 database really contains 75% of living patients as probably the database gathers patient information from previous decades.*

Thank you for pointing this out. We have reworded for accuracy.

21. - *On the different filtering strategies (Table 3), it would be interesting to see how the variant sets compare to each other and to CFTR2 (as it was done for AM in paragraph I. 211 for example)*

We apologize for the confusion. We believe Table 3 has already achieved this goal: the "CF-risk" and "non-CF-risk" correspond to the variant sets defined in CFTR2, while variant sets defined through molecular annotations were tested in Table 4 for comparison, and variant sets defined through AM are reported in Figure 3 (they depend on the AM score cutoff thus not suitable for a table presentation). We are happy to clarify further if we misunderstood this comment.

Reference

1. Sazonovs, A. et al. Large-scale sequencing identifies multiple genes and rare variants associated with Crohn's disease susceptibility. *Nat Genet* 54, 1275–1283 (2022).

2. Singh, T. et al. Rare coding variants in ten genes confer substantial risk for schizophrenia. *Nature* 604, 509–516 (2022).

3. Palmer, D. S. et al. Exome sequencing in bipolar disorder identifies AKAP11 as a risk gene shared with schizophrenia. *Nat Genet* 54, 541–547 (2022).